# Paediatric Atopic Dermatitis: The Unexpected Impact on Life with a Specific Look at the Molecular Level

**DOI:** 10.3390/ijms25094778

**Published:** 2024-04-27

**Authors:** Silvia Artusa, Giorgia Mazzuca, Giorgio Piacentini, Riccardo Castagnoli, Gian Luigi Marseglia, Angelo Pietrobelli, Luca Pecoraro

**Affiliations:** 1Pediatric Clinic, Department of Surgical Sciences, Dentistry, Pediatrics and Gynecology, University of Verona, 37126 Verona, Italy; silvia.artusa@studenti.univr.it (S.A.); giorgia.mazzuca@studenti.univr.it (G.M.);; 2Pediatric Clinic, Department of Pediatrics, Fondazione IRCCS Policlinico San Matteo, University of Pavia, 27100 Pavia, Italy

**Keywords:** atopic dermatitis, cognitive dysfunction, quality of life, school performance, anxiety

## Abstract

Atopic dermatitis (AD) is a condition with a multifactorial aetiology that affects the skin. It most often begins at preschool age and involves the skin. The disease’s main symptom is intense itching, which occurs especially at night and affects the child’s sleep, negatively impacting the quality of life of affected children and, consequently, their families. The difficulty in resting during the night leads to many problems during the day, particularly behavioural disorders and difficulties in paying attention at school, which results in learning impairment. The unexpected symptoms of AD are caused by pathophysiological processes that include many molecular pathways and inflammatory cytokines such as IL-31, IL-1, IL-2, TNF-a, and IL-6. Drawing on a comprehensive review of the literature in PubMed/MedLine, our review offers an in-depth exploration of both the psychosocial impacts of AD and the molecular processes that contribute to this disorder.

## 1. Introduction

Atopic dermatitis (AD) represents a chronic, non-infectious inflammatory disease. The primary organ involved is the skin, which has a chronic relapsing course. Atopic dermatitis is the most common disease affecting children from the first year of life. The prevalence of AD is 10.1% in Europe and 21.4% in the USA [1]. Approximately 80% of people develop atopic dermatitis during their first year of life, and the majority achieve remission during adolescence. However, some individuals may continue to experience intermittent flare-ups into adulthood [1]. The pathophysiological mechanisms underlying the development of this pathology are different. A complex interaction emerges between genetically predetermined skin barrier dysfunction, skin dysbiosis, innate and adaptive immunity dysregulation, and environmental factors [2]. More specifically, an inflammatory response is produced when environmental factors stimulate skin dysfunction, which could lead to allergic sensitization. As a result, a clinical response characterized by persistent itching and scratching, predominantly at night, develops [2]. The chronic and relapsing course of the disease, characterized by periods of remission and exacerbation, economic burden, and the whole family’s involvement in the treatment process, in association with the different possible clinical presentations in terms of severity, significantly impacts the quality of life of patients affected and of their family [2]. In particular, the physical, emotional, mental, and social health and the daily functioning of each patient can be affected [3]. As mentioned above, an inherent symptom of AD is persistent itching of the skin, arising prevalently during the night hours, interfering with daily activity and causing insomnia and sleep disorders, particularly efficiency, fragmentation, and longer latency in sleep onset. These conditions have the potential to impact a child’s neurocognitive abilities negatively and can also be related to behavioural problems such as attention-deficit/hyperactivity disorder (ADHD). Children with atopic dermatitis often perform poorly in school due to behavioural issues and learning challenges exacerbated by the physical discomfort and psychological distress associated with the condition [2]. The diagnosis of AD is primarily based on clinical symptoms and the family history of the patient in whom the condition is suspected; however, it is important to note that there is no gold-standard test available for diagnosis, underscoring the complexity of confirming the presence of atopic dermatitis [4]. Skin lesions in infants typically appear between the ages of two and six months and are characterized by papules and papulovesicles that can evolve into plaques. They usually affect the face, hands, and extensors; the scalp, neck, and trunk can also be injured. AD normally spares the diaper area. However, diaper dermatitis may occur in infants affected by AD. Flexure disease, which affects wrists, ankles, and antecubital and popliteal fossae, normally appears after 1 year of age; however, in some cases, it may be the first sign of disease [5]. Regarding the treatment of AD, the main goal is to relieve symptoms and avoid complications (both psychological and organic ones). Numerous guidelines have recently been released regarding the management of atopic dermatitis in children [4,6,7]. The approach to therapy for AD is complex, and it includes several strategies for successfully managing the disease. One of the most important tasks is to wash the skin adequately before using emollients. Emollients aid in moisturizing the skin’s surface cells and minimizing transepidermal water loss [4,7]. The application of emollients is essential, and it is recommended that they be applied to the skin twice a day [8]. Acute flares require the application of topical corticosteroids on the skin lesions. Moreover, for individuals with both mild and severe AD disease, systemic treatment with immunosuppressants and specific immunomodulating drugs may present a successful alternative treatment option, particularly in cases where conventional therapies have proven ineffective or insufficient. [4,9]. This comprehensive review aims to analyse AD’s neurocognitive consequences on children who are affected by that pathology, with a particular focus on the molecular aspects underlying the unexpected comorbidities of this disease, to raise awareness among paediatricians, families, and caregivers of children about the cognitive comorbidities and the negative implications of this pathology.

## 2. Pathophysiological Mechanisms of AD

Atopic dermatitis is characterized by intense itching, redness, and dry skin patches. The pathophysiological mechanisms underlying AD are multiple and complex. Genetic susceptibility, familial vulnerability, and predisposition have a significant role in developing this condition. These genetic factors can lead to microbial dysbiosis, immunologic dysregulation, and other crucial changes in the skin barrier [10]. Aberrant lipid metabolism and modified epidermal structural proteins, such as filaggrin and protease inhibitors, can affect the skin barrier function. These changes result in increased transepidermal water loss, elevated pH levels, susceptibility to infections, and sensitization to aeroallergens [11]. The genetic variation most substantially related to AD is filaggrin (FLG) loss of function. This protein is crucial in the differentiation of the epidermidis and in creating the skin barrier [12,13]. Functional epidermal barrier abnormalities caused by FLG mutations increase skin permeability and consequent allergy sensitization, stimulate the Th2 inflammatory response, and eventually result in asthma [12]. More than 300 FLG LoF variations are known; more than 20 are linked to AD susceptibility [14]. Atopic-dermatitis-associated genes code for immune system factors and proteins that regulate keratinocyte differentiation [15,16,17]. Certain genes linked to atopic dermatitis (AD) have been associated with pathways of the innate immune system, offering compelling evidence of the connection between the innate immune system and the disease, as well as its progression. Some of the genes identified are linked to innate immune pathways (ADAM33, MIF, MMP9, ORM2, RETN, and TLR2), which are involved in neutrophil degranulation, thereby contributing to tissue inflammation in AD [12,18,19].

Innate and adaptive immune responses are involved in developing this disease and collectively contribute to its clinical manifestation. Specifically, keratinocytes are activated in response to mechanical or inflammatory insults and produce Antimicrobial Peptides (AMPs) (cathelicidin (LL-37) and human-b-defensins 2 and 3) and pro-inflammatory cytokines. They are key players in the pathophysiology of barrier damage and the activation of the innate immune response [15,17]. AMPs play a crucial role in the restoration of a disturbed skin barrier by acting on the tight junctions [15,17]. In addition, AMPs have an autocrine effect on keratinocytes by inducing them to release pro-inflammatory molecules known as “alarmins”—in particular IL-33, IL-25, and thymic stromal lymphopoietin (TSLP). These alarmins activate innate lymphoid cells 2 (ILC2) and other dermal lymphoid cells, such as dendritic cells (DC) and Langerhans cells. These activated cells then produce IL-5 and IL-13, which in turn amplify the adaptive type 2 immune response. This self-reinforcing inflammatory cycle is triggered by the influence of AMPs on the skin barrier and keratinocytes [15,17]. The damage to the skin barrier is attributed to multiple factors, such as alterations in the synthesis of membrane proteins such as filaggrin and involucrin, changes in the metabolism of lipids and membrane ceramides, loss of tight junction integrity, and dysregulation of the keratinocyte differentiation process [15,20], but also to infectious agents (e.g., bacteria, viruses, and fungi), allergens, and mechanical trauma [20,21]. These events may be genetically determined or induced by interactions with the immune system [15,20,22]. Studies on the skin microbiome of individuals with AD have shown a reduced diversity of bacteria on healthy skin and skin with eczema. AD-affected skin is characterized by increased colonization by Staphylococcus Aureus (SA), whose microbial action is not properly balanced by other bacteria, thus becoming pathogenic and resulting in increased inflammatory interleukins (IL-4, IL-13, IL-22, and TLMP) [15,21,23]. Atopic dermatitis symptoms can also be significantly more severe due to environmental triggers such as allergies, skin irritants, infections, and physical irritants (i.e., tobacco smoke, exhaust from moving cars, extremely cold or hot weather, or humidity).

## 3. Clinical Presentation

Skin inflammation, recurrent skin infections, lack of sleep and sleep disturbances, mood changes, and a high frequency of comorbidities are some of the complex and different symptoms that children with AD experience during their lives [24].

The major distinguishing sign of atopic skin is severe xerosis, and the distribution of atopic dermatitis lesions changes with age. In particular, infants and toddlers mostly present lesions in the extremities’ extensor surfaces and the face. The skin lesions follow a variable course that includes acute and chronic phases. The initial stage of the symptoms is characterized by painful, relapsing, eczematous lesions; erythema; vesiculation; oozing; and exudation. Subsequently, lichenification and desquamation develop. In addition, these AD lesions typically can complicate infection by bacteria, viruses, and fungi. The bacteria most involved in this process is Staphylococcus Aureus. In particular, it colonizes atopic individuals’ skin densely and exacerbates the inflammatory illness [25,26].

### Itching

Atopic dermatitis is a debilitating disease. The main and most severe symptom of this disease is represented by intense and persistent itching, which is difficult to control and very disabling for children. Forty-two percent of patients experience itching for at least 18 h daily [23,27]. More than 40% of children and 60% of adults with AD experience skin pain, which can be associated with itching, scratching, skin lesions, and a neuropathic component [28,29]. The pathogenetic mechanisms underlying this symptom and the subsequent intense scratching still need to be fully understood. Itching seems to be caused by organized interactions between keratinocytes, the immune system, and sensory neurons that are not responsive to histamine [30]. By interacting with substance-specific receptors, pruritogens exacerbate itching. Unmyelinated C fibres and sparsely myelinated Aδ fibres produced from dorsal root ganglion (DRG) cell bodies mediate the itching; then, the itching signal is processed by the brain, which then causes the motor action of scratching [30]. Recent studies have shown that an increased density of nerve fibres characterizes the epidermis of children affected by AD. In addition, it has been demonstrated that a lower nerve activation threshold is characteristic of sensory nerve fibres in patients with AD [22]. This phenomenon contributes to alloknesis. This consists of the induction, by both strong itchy and non-pruritogenic stimuli, of a spot of slight, usually innocuous, itching. This condition is frequently noticed in individuals with atopic dermatitis [22]. Several itch mediators and receptors are responsible for the symptoms of AD. Th2 cells, eosinophils, neutrophils, and mast cells produce pro-inflammatory cytokines and peptides, activating pruritoceptive pathways. Th2 cells release IL-31, one of the most well-known mediators of pruritus. Research has indicated elevated IL-31 levels in AD patients’ damaged skin [30]. IL-4 and IL-13 are two more cytokines that contribute to the pathophysiology of AD and induce pruritus. The presence of IL-4 α and IL-13 α1 receptors on sensory neurons in humans and mice confirms this [30,31]. The role of these cytokines in the pathophysiology of atopic dermatitis is supported by the response to dupilumab, which interrupts IL-4 signalling, and Janus kinase (JAK) inhibitors, which result in the inhibition of IL-4 receptors [32]. Moreover, an important role is played by IL-1, whose concentrations in the serum and skin of affected individuals are very high and correlate with disease severity [22,33]. Also, keratinocytes produce pruritogen factors. One theme is alarmin TSLP, which stimulates the immune system, Th2-dependent pathway activation, and produces pro-inflammatory cytokines. The role of histamine and its receptors, particularly H1R and H4R, in the pathogenesis of pruritus in atopic dermatitis remains unclear. Histamine is commonly discharged from mast cells and basophils during a type 1 hypersensitivity response. It stimulates a specific group of sensory neurons containing histamine receptors H1R and H4R, along with transient receptor potential cation channel subfamily A member 1 (TRPV1) [34]. The concomitant inhibition of both H1R and H4R proves to be superior in alleviating both itching and inflammation compared to the individual blockade of either receptor. The clinical use of non-sedative antihistamines demonstrates limited efficacy in relieving pruritus in atopic dermatitis, indicating potential involvement of non-histaminergic pathways [30,35].

## 4. Unexpected Consequences of Atopic Dermatitis

Atopic dermatitis is related to other important comorbidities [6]. Sleep disorders represent the most disabling ones. They are very common in patients with AD and largely result from the intense itching associated with dermatitis [36]. Up to 60% of children with eczema experience sleep disturbances, which rises to 83% during flare-ups. Even in remission, eczema sufferers experience more sleep disturbances than healthy people [36]. Furthermore, the severity of this skin condition appears to impact the mood of the patients affected. Depression has been observed in both adolescents and adults affected by AD. Children and adolescents often limit their activities and also feel distressed about the appearance of their skin, which leads to avoidance of social interactions, resulting in isolation [37,38]. Moreover, an association between atopic dermatitis and behavioural disorders, such as attention deficit/hyperactivity disorder (ADHD), has been proposed, in particular in children [39,40].

### 4.1. Sleep Disorders

As mentioned above, the main symptom of atopic dermatitis is intense itching. This symptom in patients with AD is reported to become significantly more severe and painful during night hours. Itching and scratching, which disrupt falling asleep and sleep pattern continuity, increase the risk of sleep disturbances in children affected by AD [6]. Sleep problems include difficulty falling asleep, frequent nocturnal awakenings, excessive daytime sleepiness, increased sleep onset latency and awake time, and decreased sleep efficiency. Sleep problems have a significant detrimental influence on the quality of life of children with AD, as well as other members of the family, particularly parents and siblings, who also experience disrupted sleep as a result of the children’s frequent nocturnal awakenings. Interrupted sleep has been linked to lower levels of happiness, neurobehavioral and emotional impairment, hyperactivity/disattention disorder, behavioural and emotional disturbance, and poor growth in children affected [41]. Several processes contribute to sleep disturbances in children with AD. First, the stress of having a chronic disease causes acute sleeplessness, which eventually may turn chronic. Night scratching, in particular, interrupts sleep and lays the foundation for cognitive and behavioural elements that lead to insomnia and circadian rhythm disturbance as a conditioned reaction. This happens during acute periods, but sleeplessness may persist even after the acute illness is off, because sleep problems are determined by learned behavioural patterns enacted by the body even when the skin is healthy [41,42]. However, it is not just nighttime itching that causes sleep problems in AD sufferers. It has been demonstrated that scratching contributes to just 15% of nighttime awakenings in children with AD [43]. Some authors have linked the itch-gradation cycle to releasing inflammatory and pruritogenic mediators, which would further increase sleep disruption [41,44]. The circadian rhythm regulates the proper functioning of the immune system and the production of cytokines, cortisol secretion, and skin physiology. At night, increased levels of pro-inflammatory cytokines (IL-1b, IL-2, TNF-a, IFN-g, and IL-6) promote sleep, while anti-inflammatory cytokines (IL-4 and IL-10) are produced after awakening and inhibit sleep [45,46]. In patients with AD, levels of cytokines such as IL-4 are dysregulated, which may contribute to sleep disturbance [45,46]. The circadian rhythm also regulates skin barrier function. Skin blood flow is highest in the afternoon and early evening, with a second peak just before sleep onset. In addition to significant transepidermal water loss and reduced cortisol levels at sleep onset, sebum production declines at night. This may contribute to nighttime itching in AD patients [46]. Some studies have also linked atopic dermatitis to dysregulation in melatonin synthesis during the overnight period, suggesting a potential link between disrupted sleep patterns and the pathogenesis of this skin condition [46]. Chang et al. found that melatonin production during the night was higher in AD patients than in controls. In sick patients, higher levels of nocturnal melatonin were correlated with better sleep efficiency with less sleep fragmentation and milder disease [45]. Therefore, the factors underlying sleep problems in eczema sufferers are numerous and complexly intertwined (Figure 1).

### 4.2. Attention-Deficit/Hyperactivity Disorder

Attention-deficit/hyperactivity disorder (ADHD) is one of the unexpected effects of atopic dermatitis. With a 5–8% global prevalence, this is the most prevalent behavioural condition among children. Due to the significant impacts of ADHD on affected individuals and their families, such as inattention, hyperactivity, and impulsivity, these individuals often experience reduced learning abilities at school, sleep disturbances, social isolation, and, consequently, a diminished quality of life [47].

The association between AD and ADHD began to be studied in the early 1990s and has now been proven by many studies [48]. Many pathophysiological hypotheses suggest a link between ADHD and AD, but the underlying mechanisms behind this relationship have not been thoroughly examined yet. Emotional stress related to chronic illness and the subsequent intensification of TH2-related inflammatory pathways appear to involve neurotransmitters in the prefrontal cortex, which are directly involved in ADHD [48]. Many serological changes are identified in atopic dermatitis (AD), including elevated blood levels of total IgE, chemokines produced from macrophages, a chemokine that is activated and regulated by the thymus, IL-31, and a chemokine that stimulates cutaneous T cells. Evidence suggests these cytokines could trigger neuroimmunological pathways, potentially engaging cognitive systems associated with emotions and behaviour. Studies involving functional magnetic resonance imaging in humans have revealed changes in the activity of prefrontal cortex (PFC) neurons during chronic allergic responses. Any functional alterations in PFC regions have been linked with the hallmark characteristics of attention-deficit/hyperactivity disorder (ADHD) [48,49]. It is possible to conclude that individuals suffering from atopic dermatitis may be more susceptible to direct or indirect effects from the high levels of inflammatory cytokines in the brain regions linked to attention-deficit/hyperactivity disorder (ADHD), especially the prefrontal cortex (PFC) [48].

### 4.3. School Performance

Children with AD may suffer from sleep deprivation, reduced self-esteem, and impaired quality of life. All these conditions can interfere with their school attendance and performance. Certainly, the difficulties resulting from losing sleep affect both the quality and the quantity of school time. In particular, maintaining attention at school and completing homework are significantly impaired. Children with severe eczema may miss many school days because of the disease. Despite that, the correlation between atopic dermatitis and poor school performance in children suffering from AD has still been poorly investigated, underlining the need for further research to understand the impact of this skin condition on life. Lee et al. deepened data from the Korean National Health Insurance and a health screening program for newborns and children to evaluate the association between school readiness and AD [50]. Among all children born between 2008 and 2012 in Korea (average age 4.8 years), those who received an assessment of school readiness through questionnaires provided during the health screening program were enrolled in the study. The questionnaire relating to school preparation included six items: cognitive skills, social development, activity, concentration, and development of emotional and language skills. The study showed that the impact of AD is statistically associated with academic attention regardless of gender, exposure to corticosteroids and antihistamines, and age at diagnosis. In particular, the study results demonstrate more vulnerability in school preparation and activities, learning skills, and concentration in children affected by AD than in healthy ones. This study reinforces the results of existing studies in the literature [50]. Joy Wan et al. examined the prevalence of learning disabilities in a group of 2074 children diagnosed with atopic dermatitis [51]. Learning disabilities refer to disorders in reading, writing, and calculating and are associated with poor mental health, low school performance, and poor employment outcomes in adulthood. This study suggests that children with atopic dermatitis are significantly more likely than controls to have a diagnosis of learning disability, the severity of which increases as the severity of the skin disorder increases [51]. Furthermore, Sockler et al., in their longitudinal cohort study, evaluated the relationship between AD according to level of activity and severity and validated measures of general cognition (IQ scores on the GMDS, WPPSI, WISC, and WASI). However, this study did not find clinically significant associations between AD and overall cognitive function during childhood and adolescence [52]. Schmidt et al., in their nationwide cohort study conducted on 61153 Danish children, showed reduced lower and upper secondary education in children diagnosed with AD compared with children in the general population; however, the absolute differences were less than 3.5 percent. They also conducted a second analysis comparing the same AD patients with their siblings, and the estimates were less pronounced [53]. A cross-sectional study by Vittrup et al. linked data from Danish national registers by identifying three population groups between 2001 and 2019: children from lower secondary school, adolescents from upper secondary school, and male conscripts. The finding of this study was that AD, particularly the severe form, is associated with lower performance in school during childhood and lower IQs in youths. This obviously can interfere with future academic results [54]. In addition to children, studies have also been conducted on adults with AD. For children, night scratching affects daily activities such as school and homework; for adults, this involves work activities. For example, a multicentre study using TREATgermany registry data analysed the work limitations of 228 employees with atopic dermatitis using the WorkLimitations Questionnaire (WLQ). The data analysis showed that moderate-to-severe AD has a negative impact on work productivity [55].

### 4.4. Depression and Anxiety

Children suffering from atopic dermatitis experience emotional difficulties in addition to academic problems. The most important comorbidities of atopic dermatitis involve mood disorders. Among them, those most experienced by affected patients include anxiety and depression. Few studies in the literature have analysed the mental difficulties in children with atopic dermatitis. The majority of these have studied the correlations between AD and ADHD. Xie et al. deepened the possibility of mental illness in children and adolescents with AD compared to those without the disease [56]. This meta-analysis found that AD could lead to an increased risk of mental disorders among children and adolescents affected; in particular, it concluded that children with AD, compared with healthy children, are, on average, 65.2 percent more likely to develop mental disorders, including relationship problems, anxiety, depression, and increased suicide risk [56]. In a cross-sectional study conducted by Hafsia et al. in 2009 in Denmark of 9215 patients with atopic disease and, at the same time, a mental health problem, socioeconomic factors were the cause of slightly higher emotional difficulties in females than in males, while hyperactivity problems were higher in boys than in girls. Emotional difficulties increased with increasing age, but conduct problems, hyperactivity, and problems with peers were more elevated in 15-year-old adolescents. In addition, it was found that hyperactivity, emotional, and conduct problems were slightly higher in children with active symptoms than in children without active atopic disease [46]. Furthermore, a cross-sectional study by Moraes et al. investigated the association between behavioural disorders in children and adolescents and the severity of atopic dermatitis. From the analysis of a population between 6 and 17 years old, it emerged that the prevalent disorders among children and adolescents with atopic dermatitis are anxiety and depression. In addition, children with moderate and severe AD have a higher prevalence of aggressive behaviours [47]. Mental disorders can lead to negative school and employment outcomes, supporting the view that atopic dermatitis has implications for the school and work performance of those affected. The biological mechanisms behind the emotional problems linked to AD are not well understood. According to the “cytokine hypothesis of depression”, cytokines play a very important role in behavioural processes and neuroendocrine mediation leading to depressive disorders (Figure 2). Recent studies have shown that pro-inflammatory cytokines such as IL-1 and IL-6 and tumour necrosis factor (TNF-α) induce major depressive disorders in patients with chronic physical illness without a history of mental disorders [57,58]. A large amount of clinical evidence highlights the significance of the correlation between inflammation and depression in people with physical illnesses and disorders linked to elevated innate immune responses. In vulnerable patients, the breakdown of mechanisms governing sickness behaviour can happen when their inflammatory response becomes more pronounced due to an imbalance between pro-inflammatory and anti-inflammatory factors, favouring inflammation. This imbalance may manifest as the overproduction of tumour necrosis factor-alpha (TNF-α), the inadequate production of interleukin-10 (IL-10), and resistance to glucocorticoids [58].

An in vitro study conducted by Yeom et al. sought to determine how AD could result in anxiety and depression in mice by identifying signal changes in brain circuits [59]. AD-like skin lesions were induced in mice by the intradermal instillation of MC903, and then the severity of dermatitis was assessed by the evaluation of scratching, behaviour, and histopathological changes. The intradermal administration of MC903 resulted in severe scratching, significant skin inflammation with thickening of the epidermis, depressive anxiety-like behaviours, and elevated serum corticosterone levels in mice [59]. Levels of signalling molecules related to dopamine and neuronal plasticity in the nucleus accumbens, dorsal striatum, and ventral tegmental area were also determined by immunoblotting. An increase in cAMP response element-binding protein (CREB), dopamine, and cAMP-regulated phosphoprotein; an increase in brain-derived neurotrophic factor (BDNF) and ΔFosB; and a decrease in the protein expression of tyrosine hydroxylase (TH) and the dopamine D1-receptor were recorded.

## 5. Summary

Atopic dermatitis is a disorder that typically manifests in the preschool years and significantly impacts a part of the population, predominantly affecting children [1]. Therefore, it is untenable to categorize atopic dermatitis as an uncommon allergic condition [51]. The disease has several underlying causes, many of which are still unclear. Undoubtedly, atopy susceptibility resulting from a genetic and familial predisposition is a major factor in the disease’s pathophysiology, contributing significantly to its development and progression [12]. The molecular mechanisms that cause the condition are extremely complicated, with the dysregulation of pro-inflammatory cytokine release and a loss of the usual circadian rhythm of hormone release, such as cortisol and melatonin, playing critical roles in the intricate pathogenesis of the disease [15,52].

Among the symptoms of atopic dermatitis, the main one is itching, which, especially in moderate and severe cases of atopic dermatitis, is quite disabling and causes intense scratching, particularly exacerbated during nighttime and significantly impacting sleep quality and overall well-being. Patients with atopic dermatitis, therefore, spend a lot of time scratching, especially at night, and this inevitably causes a reduction in the quantity and quality of sleep [19,25]. Itching in atopic dermatitis has several mediators of dysregulated inflammation, including histamine and its receptors, IL-31, IL-1, and others [17]. Sleep disturbances in atopic dermatitis are not only due to sleep deprivation related to nighttime itching but also to dysregulation in the release of inflammatory mediators (IL-1b, IL-2, TNF-a, IFN-g, IL-6, IL-4, and IL-10) and neuroendocrine molecules such as cortisol and melatonin [36,37]. Sleep disorders lead to detrimental consequences in the daytime life of affected patients: they not only experience poor sleep quality at night but also struggle with difficulties in performing their normal activities during the day, significantly impacting their overall functioning and quality of life. Particularly in children, sleep at night adversely affects various activities, such as playing with their peers, engaging in social interactions, and, notably, their school attendance and academic performance. Children with atopic dermatitis may miss many school days during disease exacerbation, leading to poor school performance due to both the direct impact of the condition and the associated challenges in managing symptoms. In addition, for a child who may not have slept through the night due to the discomfort caused by atopic dermatitis, the time spent in school can be very tiring, with the child having difficulty paying attention and completing schoolwork both in the classroom and at home [39]. Patients with atopic dermatitis may also experience mood disorders, such as relationship problems, depression, and anxiety, which can further impair the patient’s quality of life and even increase the risk of suicide, highlighting the profound psychological impact of the condition [33]. Some pro-inflammatory cytokines such as IL-1, IL-6, and TNF-α, but also corticosteroids and dopamine, play a role in the pathogenesis of depressive disorders in individuals with atopic dermatitis. However, the mechanisms are still poorly studied [48,50]. In addition, the lockdown period during the COVID-19 outbreak further worsened the quality of life of these patients [53]. Paediatricians, schools, and families need to be aware of the unexpected comorbidities that can affect children with AD. In particular, paediatricians could monitor these aspects of the disease through apps or computerized systems that track the symptoms afflicting their patients so as not to underestimate the unexpected complications of the disease and constantly stay connected with the patient by assessing the worsening or improvement of symptoms.

## 6. Conclusions

Despite the lack of studies regarding these topics, a relationship between atopic dermatitis, sleep disorders, poor school performance, and behaviour disorders exists, suggesting a complex interplay among these factors in impacting the lives of affected individuals. Underlying this relationship are several complex molecular and neuroendocrine factors that have not yet been fully elucidated or comprehensively studied, highlighting the need for further research to better understand the intricate mechanisms at play. Therefore, AD has an important impact on the quality of life and psychological health of children affected and their families. This underlines the importance of multi-disciplinary teams, including the family paediatrician, teachers, professors, and mental health professionals, taking care of children with atopic dermatitis.

## Figures and Tables

**Figure 1 ijms-25-04778-f001:**
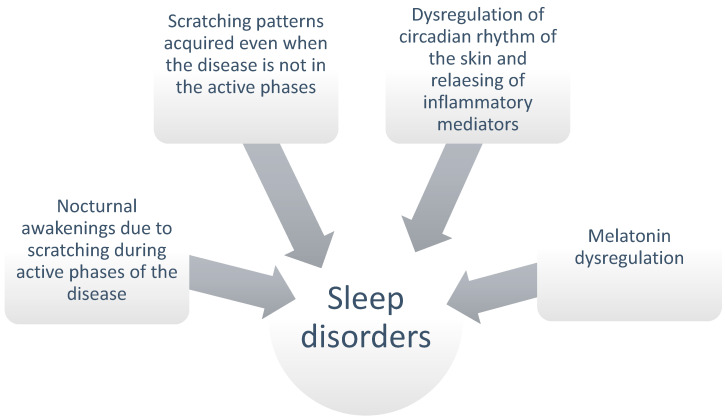
Multifactorial etiopathogenesis of sleep disorders in patients with atopic dermatitis.

**Figure 2 ijms-25-04778-f002:**
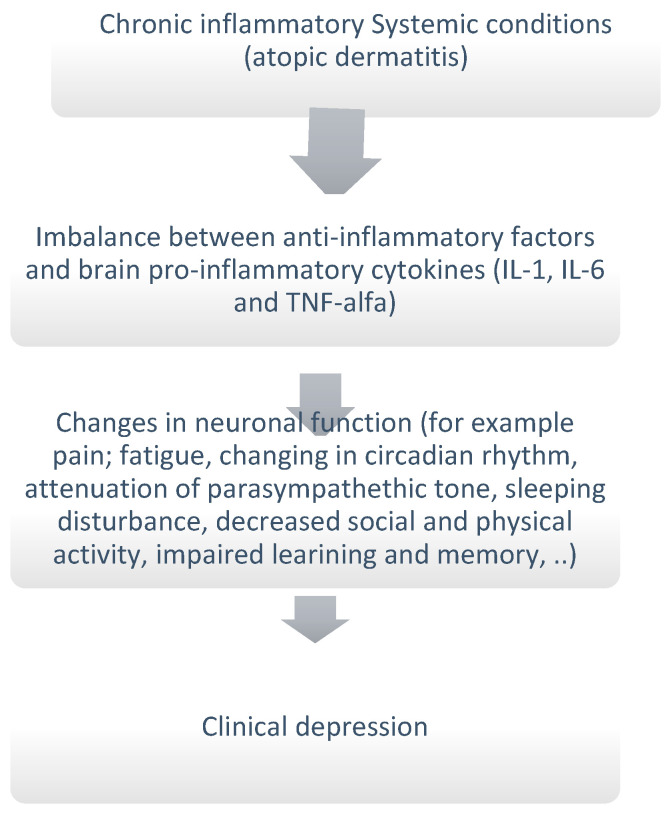
Molecular mechanism underlying depression in patients with systemic inflammatory conditions such as atopic dermatitis.

## Data Availability

Not applicable.

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
