# Peer review of "Paediatric Atopic Dermatitis: The Unexpected Impact on Life with a Specific Look at the Molecular Level"

_ijms, 2024, doi:10.3390/ijms25094778_

Round 1

Reviewer 1 Report

Comments and Suggestions for Authors

This article provides an overview of the multiple effects of atopic dermatitis (AD) on the daily lives of affected children. It highlights the significant psychosocial consequences, including sleep disturbances, behavioral problems and learning difficulties, which can affect the quality of life of children and their families. Overall, this review is interesting, but the manuscript needs careful revision and intensive editing. I strongly recommend a second round of peer review.

Here are some comments/suggestions/queries for the authors to consider for improvement

Title and abstract

-        The given title "Pediatric Atopic dermatitis: the unexpected impact of life with a specific look at the molecular level" is not entirely consistent with the abstract presented. The abstract focuses primarily on the psychosocial consequences of AD in children, such as sleep disturbances, behavioral problems, and learning difficulties. The aspects of the disease at the molecular level implied in the title are not addressed.

-          The authors can introduce the abbreviation "AD" when first mentioning "atopic dermatitis" in the abstract, and then use the abbreviation consistently throughout the rest of the text.

-          The authors should include a section in the abstract that briefly outlines the key molecular mechanisms and findings related to the condition.

Introduction

-          Line 31-32. The prevalence ranges reported for Europe and the USA

-          Lines 32-33. The authors stated that “ .. seems to be exponentially increasing during the last few years [1], but the reference does not appear to make this claim. The authors should remove or qualify this statement if they do not have sufficient evidence to support it.

-          The statements in line 60 and line 64 appear redundant as they both convey the same information that the diagnosis of ADis primarily based on clinical assessment and history. The authors should avoid redundancies throughout the manuscript and improve the flow of the text.

-          Line 82. Based on the information provided in reference 8, the statement that “It has recently been found that it is recommended….” is not accurate. The reference actually recommends twice-daily application of moisturizers, not a single daily application.

-          Line 89-92 the authors stated that their review aims to analyze the neurocognitive impacts of AD on children, in order to raise awareness among pediatricians, families, and caregivers. However, the abstract states the review aims to analyze the psychosocial consequences of AD, again to raise awareness among pediatricians and families. These two statements do not fully align, as neurocognitive consequences and psychosocial consequences are distinct aspects that may require different analytical approaches and have different implications. The authors should ensure the objectives and scope of the review are consistently described in both the introduction and abstract.

-          Line 109. The authors should spell out the full term and provide the abbreviation in parentheses when introducing an acronym or abbreviation for the first time throughout the text. "Antimicrobial Peptides (AMPs)"

-          Lines 107-110. Reference is missing

-          Lines 112-119. The authors have clearly copied and pasted this content from another review article (PMID: 36359220) without properly citing the original source. The authors' actions in this case constitute plagiarism.

-          Line 119: The wording "The damage to the skin barrier is determined by" is not quite correct. The wording "The damage to the skin barrier is caused by or attributed to" would be more correct. Furthermore, the authors have only listed four factors that contribute to skin barrier impairment; what about infectious agents (e.g. bacteria, viruses, fungi), allergens and mechanical trauma? The authors need to substantiate their information here and throughout the manuscript with appropriate references.

-          Line 94. The authors can address the specific genetic and immunologic factors that contribute to the development and progression of AD. They can also discuss the interaction of the various pathophysiological mechanisms and how they collectively contribute to the clinical manifestations of the disease.

-          Lines 169-171. Reference is missing

-          Line 158. Correct “patients experience it” to “42% of patients experience itch”.  This is also copied and pasted from PMID: 38108679.

-          Line 181. The authors should prioritize citing the original research studies over review articles when supporting their claims. For example, “Reference 26” is a review article not a research article.

-          Lines 210-211. Reference is missing

-          Line 231. The authors have made a mistake in their statement. Instead of referring to children with "Alzheimer's disease", they should have mentioned children with AD"

-          Line 237. The full stop should come after the reference [37], not before it.

-          Lines 237-240. The authors use of the phrase "such as AD" is redundant and unnecessary in this context, as they have already established that they are discussing sleep disturbances in children with AD.

-          Lines 251-256. References are missing.

-          Fig. 1. Correct “Disregulation” to “Dysregulation”. Also, the factors "Dysregulation of circadian rhythm of the skin" and "Dysregulation of circadian rhythm of release of inflammatory mediators" do appear to be closely related and could be combined into a single factor. The authors could expand the figure to include potential interventions or strategies that could help mitigate the impact of these factors on sleep quality in AD patients.

-          Lines 284-286. The authors' statement that "Many pathophysiological hypotheses support the correlation between ADHD and AD" is not entirely accurate. Hypotheses can suggest or propose potential links, but they do not directly support a correlation. The association between AD and ADHD has been observed in epidemiological studies, but the underlying mechanisms behind this relationship have not been thoroughly examined. The authors should rephrase their statement to reflect this more accurately.

-          Line 406. Fig. 2 and legend do not adequately explain the "molecular mechanism" underlying the association between atopic dermatitis and depression. The figure demonstrates a simplified pathway where chronic inflammatory conditions, such as AD, lead to increased production of pro-inflammatory cytokines (IL-1, IL-6, TNF-alpha), which then result in depression. However, the molecular mechanisms linking these factors are not clearly elucidated.

-          Line 426. The authors have not introduced any new insights or perspectives in this section and have largely restated the points made earlier in the text.

-        The writing (precision of expression; English; references) should be much better.

Comments on the Quality of English Language

Editing of English language required

Author Response

Title and abstract

-        The given title "Pediatric Atopic dermatitis: the unexpected impact of life with a specific look at the molecular level" is not entirely consistent with the abstract presented. The abstract focuses primarily on the psychosocial consequences of AD in children, such as sleep disturbances, behavioral problems, and learning difficulties. The aspects of the disease at the molecular level implied in the title are not addressed.

-          The authors can introduce the abbreviation "AD" when first mentioning "atopic dermatitis" in the abstract, and then use the abbreviation consistently throughout the rest of the text.

-          The authors should include a section in the abstract that briefly outlines the key molecular mechanisms and findings related to the condition.

We sincerely thank the reviewer, we have implemented the abstract according to your suggestions.

Introduction

-          Line 31-32. The prevalence ranges reported for Europe and the USA

We thank the reviewer for the suggestion. We have reported the additions requested in the introduction section about the prevalence of AD.

-          Lines 32-33. The authors stated that “ .. seems to be exponentially increasing during the last few years [1], but the reference does not appear to make this claim. The authors should remove or qualify this statement if they do not have sufficient evidence to support it.

We thank the reviewer for the suggestion. We have removed the statement.

-          The statements in line 60 and line 64 appear redundant as they both convey the same information that the diagnosis of ADis primarily based on clinical assessment and history. The authors should avoid redundancies throughout the manuscript and improve the flow of the text.

We sincerely thank the reviewer. We have removed the redundant statement.

-          Line 82. Based on the information provided in reference 8, the statement that “It has recently been found that it is recommended….” is not accurate. The reference actually recommends twice-daily application of moisturizers, not a single daily application.

We kindly thank the reviewer. We have rectified the sentence with the correct information.

-          Line 89-92 the authors stated that their review aims to analyze the neurocognitive impacts of AD on children, in order to raise awareness among pediatricians, families, and caregivers. However, the abstract states the review aims to analyze the psychosocial consequences of AD, again to raise awareness among pediatricians and families. These two statements do not fully align, as neurocognitive consequences and psychosocial consequences are distinct aspects that may require different analytical approaches and have different implications. The authors should ensure the objectives and scope of the review are consistently described in both the introduction and abstract.

We sincerely thank the reviewer for this suggestion. We have adjusted the objectives and scope of the review in both the introduction and abstract.  

-          Line 109. The authors should spell out the full term and provide the abbreviation in parentheses when introducing an acronym or abbreviation for the first time throughout the text. "Antimicrobial Peptides (AMPs)"

We kindly thank the reviewer. We have spelled out the full term and then we have provided the abbreviation for AMPs.

-          Lines 107-110. Reference is missing

We thank the reviewer for the suggestion. We have added the reference to line 107-110.

-          Lines 112-119. The authors have clearly copied and pasted this content from another review article (PMID: 36359220) without properly citing the original source. The authors' actions in this case constitute plagiarism.

We acknowledge that there was an oversight in properly citing the original source for the content mentioned. This was not intentional, and we apologize for it. We have promptly corrected this by providing the appropriate citation to the original article (PMID: 36359220) in our manuscript. Thank you for bringing this to our attention.

-          Line 119: The wording "The damage to the skin barrier is determined by" is not quite correct. The wording "The damage to the skin barrier is caused by or attributed to" would be more correct. Furthermore, the authors have only listed four factors that contribute to skin barrier impairment; what about infectious agents (e.g. bacteria, viruses, fungi), allergens and mechanical trauma? The authors need to substantiate their information here and throughout the manuscript with appropriate references.

- Line 94. The authors can address the specific genetic and immunologic factors that contribute to the development and progression of AD. They can also discuss the interaction of the various pathophysiological mechanisms and how they collectively contribute to the clinical manifestations of the disease.

We sincerely thank the reviewer. We have updated the line 119 and 94 with more informations and appropriate references.

-          Lines 169-171. Reference is missing

We thank the reviewer for the suggestion. We have added the reference to line 169-181.

-          Line 158. Correct “patients experience it” to “42% of patients experience itch”.  This is also copied and pasted from PMID: 38108679.

We thank the reviewer for the suggestion. We have corrected the sentence. We apologize again for missing the citation of original article PMID: 38108679, we have added the reference.

-          Line 181. The authors should prioritize citing the original research studies over review articles when supporting their claims. For example, “Reference 26” is a review article not a research article.

We sincerely thank the reviewer. We have changed the reference with the correct research articles.

-          Lines 210-211. Reference is missing

We thank the reviewer for the suggestion. We have added the reference to line 210-211.

-          Line 231. The authors have made a mistake in their statement. Instead of referring to children with "Alzheimer's disease", they should have mentioned children with AD"

We thank you the reviewer for the suggestion. We have corrected the sentence.

-          Line 237. The full stop should come after the reference [37], not before it.

We thank you the reviewer for the suggestion. We have corrected the sentence.

-          Lines 237-240. The authors use of the phrase "such as AD" is redundant and unnecessary in this context, as they have already established that they are discussing sleep disturbances in children with AD.

We sincerely thank the reviewer. We have removed the redundant word.

-          Lines 251-256. References are missing.

We thank the reviewer for the suggestion. We have added the reference to line 261-256.

-          Fig. 1. Correct “Disregulation” to “Dysregulation”. Also, the factors "Dysregulation of circadian rhythm of the skin" and "Dysregulation of circadian rhythm of release of inflammatory mediators" do appear to be closely related and could be combined into a single factor. The authors could expand the figure to include potential interventions or strategies that could help mitigate the impact of these factors on sleep quality in AD patients.

Thanks for your suggestion. We corrected “Disregulation” to “Dysregulation”. We combined the factors "Dysregulation of circadian rhythm of the skin" and "Dysregulation of circadian rhythm of release of inflammatory mediators". About the possibility to expand the figure, we thank the reviewer for his/her suggestion. Anyway, we propose not to expand the figure.

-          Lines 284-286. The authors' statement that "Many pathophysiological hypotheses support the correlation between ADHD and AD" is not entirely accurate. Hypotheses can suggest or propose potential links, but they do not directly support a correlation. The association between AD and ADHD has been observed in epidemiological studies, but the underlying mechanisms behind this relationship have not been thoroughly examined. The authors should rephrase their statement to reflect this more accurately.

We thank you the reviewer for the suggestion. We have rephrased the statement as he recommended us.

-          Line 406. Fig. 2 and legend do not adequately explain the "molecular mechanism" underlying the association between atopic dermatitis and depression. The figure demonstrates a simplified pathway where chronic inflammatory conditions, such as AD, lead to increased production of pro-inflammatory cytokines (IL-1, IL-6, TNF-alpha), which then result in depression. However, the molecular mechanisms linking these factors are not clearly elucidated.

We sincerely thank the reviewer. We have updated fig.2 with more informations and appropriate references.

-          Line 426. The authors have not introduced any new insights or perspectives in this section and have largely restated the points made earlier in the text.

We thank you the reviewer for the comment. We reorganized the section 4.4 and we have removed the redundant final sentence.  

-        The writing (precision of expression; English; references) should be much better. Comments on the Quality of English Language: Editing of English language required.

We warmly thank the reviewer for the comment. We have edited our work by correcting grammatical mistakes and making the text as fluent as possible.

Reviewer 2 Report

Comments and Suggestions for Authors

The work titled "Pediatric Atopic dermatitis: the unexpected impact of life with a specific look at the molecular level" presented as a proposal for publication in IJMS is a well-written work that contains all the necessary elements. It is written clearly for the reader and explains thoroughly all issues related to the problem of people affected by AD and their immediate environment. Personally, this work lacks proposals/plans of what could be done, what steps to take that would help in AD therapy. However, this is only a suggestion that does not reduce the value of the presented article.

Author Response

The work titled "Pediatric Atopic dermatitis: the unexpected impact of life with a specific look at the molecular level" presented as a proposal for publication in IJMS is a well-written work that contains all the necessary elements. It is written clearly for the reader and explains thoroughly all issues related to the problem of people affected by AD and their immediate environment. Personally, this work lacks proposals/plans of what could be done, what steps to take that would help in AD therapy. However, this is only a suggestion that does not reduce the value of the presented article.

We sincerely thank the reviewer for comprehending the significance of our comprehensive review. We think pediatricians could stay in touch with the patient through computerized tools so that they are always up to date on the clinical progress of the condition. We added this concept to the article.

Round 2

Reviewer 1 Report

Comments and Suggestions for Authors

The authors have addressed most of my concerns with the original manuscript. Here are some other suggestions to be consider.

Abstract:

Line 12. What AD stand for? I recommended to the authors in my previous report to introduce the abbreviation "AD" when first mentioning "atopic dermatitis" in the abstract, and then use the abbreviation consistently throughout the rest of the text. This comment is also applies for the other sections of the manuscript.

Line 21-22. The current statement "Our comprehensive review, starting with an analysis of the literature published in PubMed/MedLine, aims …." could potentially give the impression that your review is a systematic one. However, as a narrative review, your manuscript takes a less formal approach than systematic reviews and does not require the presentation of rigorous aspects such as reporting methodology, search terms, databases used, and inclusion and exclusion criteria. Therefore, I propose revising the sentence to something like this: "Drawing on a comprehensive review of the literature in PubMed/MedLine, our review offers an in-depth exploration of both the psychosocial impacts of AD and the molecular processes that contribute to this disorder"

Lines 45-49. Provide reference.

Lines 127-134. Can be rephrased as follows: “AMPs play a crucial role in the restoration of a disturbed skin barrier by acting on the tight junctions [15], [17]. In addition, AMPs have an autocrine effect on keratinocytes by inducing them to release pro-inflammatory molecules known as "alarmins" – in particular IL-33, IL-25 and thymic stromal lymphopoietin (TSLP). These alarmins activate innate lymphoid cells 2 (ILC2) and other dermal lymphoid cells, such as dendritic cells (DC) and Langerhans cells. These activated cells then produce IL-5 and IL-13, which in turn amplify the adaptive type 2 immune response. This self-reinforcing inflammatory cycle is triggered by the influence of AMPs on the skin barrier and keratinocytes.

Line 138. References should be placed after the word "process"

 Fig. 1. Correct “Disregulation” to “Dysregulation”.

Line 451. The text you provided seems to be a summary or recap of information previously discussed in the review. A discussion in a literature review typically involves synthesizing and comparing the findings from the reviewed studies. The current content does not fulfill this purpose, it may need to be revised or restructured to better reflect the discussion of the literature. Alternatively, the subheading “Discussion” can be changed to “Summary” and the text retained.

Lines 501-502. The authors need to maintain consistency in their terminology “Pediatricians” and “paediatricians”.

Author Response

The authors have addressed most of my concerns with the original manuscript. Here are some other suggestions to be consider.

Abstract:

Line 12. What AD stand for? I recommended to the authors in my previous report to introduce the abbreviation "AD" when first mentioning "atopic dermatitis" in the abstract, and then use the abbreviation consistently throughout the rest of the text. This comment is also applies for the other sections of the manuscript.

We thank the reviewer for the comment. Now we have introduced the abbreviation 'AD' the first time we mentioned atopic dermatitis.

Line 21-22. The current statement "Our comprehensive review, starting with an analysis of the literature published in PubMed/MedLine, aims …." could potentially give the impression that your review is a systematic one. However, as a narrative review, your manuscript takes a less formal approach than systematic reviews and does not require the presentation of rigorous aspects such as reporting methodology, search terms, databases used, and inclusion and exclusion criteria. Therefore, I propose revising the sentence to something like this: "Drawing on a comprehensive review of the literature in PubMed/MedLine, our review offers an in-depth exploration of both the psychosocial impacts of AD and the molecular processes that contribute to this disorder"

We thank the reviewer very much for the suggestion; we have made the change to the sentence as suggested.

Lines 45-49. Provide reference.

We thank the reviewer for the suggestion. We have added the reference to line 45-49.

Lines 127-134. Can be rephrased as follows: “AMPs play a crucial role in the restoration of a disturbed skin barrier by acting on the tight junctions [15], [17]. In addition, AMPs have an autocrine effect on keratinocytes by inducing them to release pro-inflammatory molecules known as "alarmins" – in particular IL-33, IL-25 and thymic stromal lymphopoietin (TSLP). These alarmins activate innate lymphoid cells 2 (ILC2) and other dermal lymphoid cells, such as dendritic cells (DC) and Langerhans cells. These activated cells then produce IL-5 and IL-13, which in turn amplify the adaptive type 2 immune response. This self-reinforcing inflammatory cycle is triggered by the influence of AMPs on the skin barrier and keratinocytes.

We sincerely thank the reviewer. We have updated the lines 127-134 as suggested.

Line 138. References should be placed after the word "process"

We thank the reviewer for the suggestion. We have changed the place of the reference in the text.  

 Fig. 1. Correct “Disregulation” to “Dysregulation”.

We thank very much the reviewr for the comment. We have corrected the word “dysregulation”.

Line 451. The text you provided seems to be a summary or recap of information previously discussed in the review. A discussion in a literature review typically involves synthesizing and comparing the findings from the reviewed studies. The current content does not fulfill this purpose, it may need to be revised or restructured to better reflect the discussion of the literature. Alternatively, the subheading “Discussion” can be changed to “Summary” and the text retained.

We sincerely thank the reviewer for the suggestion. We have changed the title of the paragraph as suggested.

Lines 501-502. The authors need to maintain consistency in their terminology “Pediatricians” and “paediatricians”.

We thank very much the reviewer for the comment. We have corrected the word.